# 17⍺-Estradiol Protects against HIV-1 Tat-Induced Endolysosome Dysfunction and Dendritic Impairments in Neurons

**DOI:** 10.3390/cells12050813

**Published:** 2023-03-06

**Authors:** Gaurav Datta, Nicole M. Miller, Xuesong Chen

**Affiliations:** Department of Biomedical Sciences, University of North Dakota School of Medicine and Health Sciences, Grand Forks, ND 58203, USA

**Keywords:** HIV-1 Tat, endolysosomes, 17α-estradiol, estrogen receptors alpha, dendritic spine

## Abstract

HIV-1 Tat continues to play an important role in the development of HIV-associated neurocognitive disorders (HAND), which persist in 15–55% of people living with HIV even with virological control. In the brain, Tat is present on neurons, where Tat exerts direct neuronal damaging effects by, at least in part, disrupting endolysosome functions, a pathological feature present in HAND. In this study, we determined the protective effects of 17α-estradiol (17αE2), the predominant form of estrogen in the brain, against Tat-induced endolysosome dysfunction and dendritic impairment in primary cultured hippocampal neurons. We demonstrated that pre-treatment with 17αE2 protected against Tat-induced endolysosome dysfunction and reduction in dendritic spine density. Estrogen receptor alpha (ERα) knockdown impairs the ability of 17αE2 to protect against Tat-induced endolysosome dysfunction and reduction in dendritic spine density. Furthermore, over-expressing an ERα mutant that fails to localize on endolysosomes impairs 17αE2′s protective effects against Tat-induced endolysosome dysfunction and reduction in dendritic spine density. Our findings demonstrate that 17αE2 protects against Tat-induced neuronal injury via a novel ERα-mediated and endolysosome-dependent pathway, and such a finding might lead to the development of novel adjunct therapeutics against HAND.

## 1. Introduction

The effective suppression of HIV-1 replication by combined antiretroviral therapy (ART) has significantly decreased mortality rates and improved the lifespan and quality of life of people living with HIV (PLWH). However, HIV-associated neurocognitive disorders (HAND) persist in 15–55% of PLWH even with ART [1,2,3]. One of the key pathological features of HAND that correlate closely with neurocognitive impairment is synaptodendritic impairments that occur in discrete brain regions such as the prefrontal cortex and hippocampus [4,5,6,7,8,9,10,11]. Although the underlying mechanisms of synaptodendritic impairments in HAND remain elusive in the ART era, various HIV-related factors are involved, including the presence of low levels of HIV-1 from the reactivation of latent infected cells [12], HIV-1 viral proteins [13,14,15,16], ART drugs [17], and substance use disorders [18,19,20]. Reversible synaptodendritic impairment resulting from these HIV-related factors contribute to the development of mild cognitive impairment in PLWH [21,22,23,24,25]. 

At the cellular level, many of these HIV-1 related factors could induce endolysosome dysfunction, a pathological feature that is present in the post-mortem brains of HAND patients [26,27,28,29]. As acidic organelles that are responsible for degradation and membrane trafficking, endolysosomes are especially critical for neuronal function because neurons are long-lived post-mitotic cells with extensive processes. As long-lived post-mitotic cells, neurons cannot remove non-degraded or partially degraded and potentially toxic constituents through cell division, and thus quality control over a large volume of cytoplasm and membrane relies on the efficient proteolytic degradation capability of endolysosomes [30,31]. Furthermore, endolysosomes in neurons serve as a robust and dynamic intracellular vesicular trafficking machinery, which is required for establishing and maintaining the axonal and somatodendritic membrane domains. As such, endolysosomes play an important role in modulating dendritic and synaptic plasticity [32,33,34,35,36,37,38]. Thus, endolysosome dysfunction could lead to synaptodendritic impairments. Conversely, enhancing endolysosome function and/or alleviating endolysosome dysfunction represents a promising therapeutic strategy against HAND.

Among various HIV-related factors, HIV-1 Tat has been consistently demonstrated to play an important role in the pathogenesis of HAND. As an essential protein for HIV-1 viral transcription, Tat can be actively secreted from infected cells [39,40,41,42]. Significantly, current anti-HIV strategies do not block the secretion of Tat [43], and brain levels of Tat remain elevated despite virological control [44,45]. In the brain, Tat has been detected on neurons [46,47], where Tat can exert neurotoxic effects directly [48,49] and induce synaptodendritic impairment [50,51,52,53,54,55]. Such Tat-mediated direct neuronal damaging effects depend, at least in part, on the internalization of Tat into endolysosomes [54,55,56,57,58,59,60,61]. We have shown that Tat induces endolysosome dysfunction in neurons [60,61]. Thus, Tat-induced endolysosome dysfunction could contribute to the direct neuronal damaging effect of Tat. 

Based on our recent findings that 17α-estradiol (17αE2), the predominant form of estrogen in the brain, exerts enhancing effects on endolysosome function and dendritic spine formation via estrogen receptor alpha (ERα) [62], we determined the extent to which 17αE2 protects against HIV-1 Tat-induced endolysosome dysfunction and dendritic impairment in primary cultured hippocampal neurons.

## 2. Materials and Methods

Cell Cultures: ERα-expressing immortalized mouse E-18 neurons CLU199 (Cellutions Biosystems, Cedarlane, ON, Canada) were grown and maintained in 1× DMEM with 25 mM glucose, 10% fetal bovine serum (FBS), and 1% penicillin/streptomycin and maintained in a 37 °C incubator with 5% CO_2_. Cells from passages 3–7 were used for all experiments in this study. Primary cultured mouse hippocampal neurons were obtained from BrainBits LLC (C57EHP, Springfield, IL, USA), and the neurons were grown as per the manufacturer’s instructions. The neurons, at a density of 100,000 cells per well, were plated on either poly-D-lysine-coated 12 mm coverslips (GG-12-PDL, Neuvitro Corporation, Vancouver, WA, Canada) or on ploy-D-lysine-coated 35 mm glass bottom dishes (P35GC-0-10-C, MatTek Life Sciences, Ashland, MA, USA). NbActiv medium (BrainBits) was used for both the plating and maintenance, and half the medium was replaced with new medium every 3–4 days. The neurons at DIV 12–17 were used for experiments. 

Dendritic Spine Live Imaging: Mouse hippocampal neurons expressing cytoplasmic GFP (BacMam GFP, B10383, Thermo Fisher, Waltham, MA, USA) were used, and images were acquired with a Zeiss LSM800 Confocal microscope using the 63X objective. The neurons were treated with recombinant HIV-1 Tat (ImmunoDx, Woburn, MA, USA) and/or 17αE2 (10 nM, Tocris, Minneapolis, MN, USA), according to experimental design, with heat-inactivated Tat (95 °C for 1 h) used as a control. Images were acquired at 1 min intervals for 10 min with a z-stack set at 0.5 μm. From the acquired confocal images at 0 (t-0) and 10 min (t-10), the dendritic spines were reconstructed in 3D with Imaris 9.5. using the filaments module. The percentage of spines lost/gained over a 10 min period was plotted. For each treatment, at least 7–10 neurons were imaged, and more than 5000 spines were analyzed. Experiments were repeated three times with different cultures of neurons. Spine classification criteria published in an earlier study [63] were used.

Immunostaining: Following treatment as per the experimental design, the neurons were fixed with 4% PFA at RT for 20 min, and 0.05–0.1% Triton X-100 treatment for 10 min was used to permeabilize the membranes. Following blocking (3% BSA with 1% normal goat or donkey serum in PBS) for 1 h, the neurons were incubated with MAP2 antibody (1:500, ab32454, Abcam, Waltham, MA, USA) overnight at 4 °C. After washing, the neurons were incubated with Alexa Fluor 488 goat anti-rabbit (1:500, Thermo Fisher) for 1 h. The cells on the coverslips were mounted onto microscope slides (Fisher Scientific) and the cells in the 35 mm dishes were used directly with ProLong Gold Antifade (P36930, Thermo Fisher). Cells stained only with primary antibodies (background controls) or only with secondary antibodies were used as controls for eliminating autofluorescence and bleed-through between channels.

Dendritic Spine Density Measurement in Fixed Neurons: Neurons stained with MAP2 were used for assessing dendritic morphology and length. Neurons stained with phalloidin were used for the measurement of dendritic spines as phalloidin labels F-actin, which is enriched at spine heads and serves as a marker for dendritic spines. Following the acquisition of confocal images under a 63X objective using a Zeiss LSM800 confocal microscope, the dendrites and spines were reconstructed in 3D with Imaris 9.5 using the filaments module. Multiple dendrites from different neurons were reconstructed and the spines were classified using a previously defined criterion. The number of dendritic spines/10 μm of primary and secondary neurites was calculated and served as a measure of dendritic spine density.

Endolysosome pH Measurement: The luminal pH of the endolysosomes in the CLU199 cells was measured ratiometrically using a pH sensitive and pH insensitive dextran, as described previously [64]. CLU199 cells plated on 35 mm glass bottom dishes were incubated with a pH insensitive dextran Texas Red (10 μg/mL, D1863, Thermo Fisher) and a pH sensitive pHrodo Green dextran (10 μg/mL, P35368, Thermo Fisher). Following a 24 h incubation period, the dextran-containing medium was replaced with warm Hibernate E Low Fluorescence Medium (HELF, Brainbits, Springfield, Illinois, USA) for imaging under a Zeiss LSM800 confocal microscope. Images were acquired with z-stacks at 1 μm intervals and fluorescence emission at 533 nm for Green dextran and 615 nm for Texas Red dextran. For pH calibration, a standard curve was determined in another set of cells incubated with buffers of different pH with the addition of 10 μM nigericin and 20 μM Monensin in Hibernate E Flow Fluorescence (HELF) Medium, which was performed as per the intracellular pH calibration kit (P35379, Thermo Fisher) instructions. Using this standard curve, the fluorescence ratio at 615/533 was converted to pH. In these experiments, a 40X objective was used, a total of 5 fields with at least 5–10 cells per field were imaged, and the experiments repeated independently three times. 

Active Cathepsin D Staining: Active cathepsin D was identified in cells stained with BODIPY-FL Pepstatin A (P12271, Thermo Fisher), and total endolysosomes were identified with LysoTracker Red DND-99. Briefly, the cells were incubated with LysoTracker Red DND-99 (10 nM) and with BODIPY-FL Pepstatin A (1 μM) for 30 min at 37 °C. Following washing, fresh, warm Hibernate E low fluorescence (HELF) medium was added for imaging under the 63X objective of a Zeiss LSM 800 confocal microscope using 0.5 μm z-stack intervals. Twenty-five to thirty cells per treatment group were imaged, and the experiments were repeated independently three times. Total endolysosomes (LysoTracker Red) and active cathepsin D positive endolysosomes (BODIPY-FL Pepstatin A) were reconstructed as spots using Imaris 9.5 software, and the percentage of active cathepsin D positive endolysosomes vs. total endolysosomes was calculated. 

ERα over-expression and siRNA knockdown: ERα-HA and ERα C451A-HA cloned into pCMV6-AC-3HA vectors were obtained from Origene Technologies (Rockville, MD, USA). Lipofectamine 2000 transfection Reagent (11668019, ThermoFisher) was used for transient transfections, and cells plated on either 35 mm glass bottom dishes or 12 mm coverslips were transfected with 1–2 µg of plasmid DNA/well in Opti-MEM Reduced Serum medium (31985062, Thermo Fisher) for 48 h. On-Target plus mouse Esr1 (Entrez Gene ID-13982) siRNA-SMART pool (Horizon Discovery Biosciences Limited) was used for the siRNA knockdown of ERα. The following target sequences were used: CCUACUACCUGGAGAACGA, GAAAGGCGGCAUACGGAAA, GUCCAGCAGUAACGAGAAA, and GGGCUAAAUCUUGGUAACA. Accell1 transfection media (B-005000, Dharmacon/Horizon Discovery Biosciences Limited, Waterbeach, Cambridge, UK) and DharmaFECT 1 (T-2001-02, Dharmacon) were used as transfection reagents, and siRNA at a concentration of 50 nM was used. Following 48 h of siRNA treatment, the transfection efficiency was determined using immunoblotting. 

Immunoblotting: CLU199 cells or primary mouse hippocampal neurons plated on Poly D-lysine-coated 6-well plates at 1 × 106/well or 0.5 × 10^6^/well, respectively, were used for immunoblotting. Following washing and harvesting, the cells were lysed in ice-cold 1 × IP lysis buffer (Thermo Fisher) with Protease Inhibitor Cocktail (Pierce) for 30 min on ice. Cell lysates were cleared with centrifugation (13,000× *g* for 10 min at 4 °C), and the protein concentrations of the collected supernatants were determined using Precision Red Advanced Assays (Cytoskeleton Inc., Denver, CO, USA). The proteins (50 μg) were separated with 4–20% SDS-PAGE gel and transferred to PVDF membranes (Thermo Fisher). The membranes were kept in blocking (LiCor) for 1 h, followed by overnight incubation with N-terminal anti-ERα antibody (1:500, sc-8002 Santa Cruz) at 4 °C, with GAPDH as a loading control (1:2000, ab8245, Abcam). Following washing with TBS, the membranes were incubated with LiCor IR secondary antibodies (1:5000). After washing, the blots were imaged, and the densitometries of the blots were quantified and analyzed using Li-Cor Odyssey Fc Imager. 

Statistical Analysis: All data were expressed as mean ± SEM. For each experiment, the “n” was specified in the figure legend. GraphPad Prism 9.0 (GraphPad Software, Inc., Boston, MA, USA) was used for data analysis and the preparation of all the graphs. For the statistical analysis, Student’s *t*-tests (two-tailed) or one-way or two-way ANOVA with Tukey’s post-hoc tests were used to calculate the statistical significance between the groups, and an alpha of 0.05 was used as the cutoff for significance. 

## 3. Results

### 3.1. Tat Induces Dendritic Spine Impairment and Endolysosome Dysfunction

As a secreted HIV-1 protein, Tat can enter all CNS cells via endocytosis by interacting with cell surface receptors or proteins [54,55,56,57]. Tat has been detected on neurons [46], astrocytes [46], and microglia [47] in the brain, where Tat can exert neurotoxic effects directly [48,49] and indirectly [65,66]. Here, we determined the extent to which Tat affects dendritic length and dendritic spines in primary hippocampal neurons. Morphological changes to the structure or configuration of dendrites were assessed with MAP2 staining, and morphological changes in F-actin-laden dendritic spines were assessed with Alexa Fluor-488 Phalloidin (Figure 1A) in neurons treated with different concentrations of HIV-1 Tat (10 nM, 50 nM, and 100 nM) for 48 h. Although Tat did not significantly reduce dendritic length, Tat (100 nM) induced a significant decrease in dendritic spine density (Figure 1B), a finding that is consistent with those of other studies [50,51,52,53]. 

Such Tat-mediated direct neuronal damaging effects depend, in part, on the internalization of Tat into endolysosomes [54,55,56,57,58,59,60,61]. Thus, internalized Tat can disturb endolysosome function directly. Indeed, we have shown that Tat induces endolysosome dysfunction in rat neurons [60,61]. Here, we determined the effects of Tat on endolysosome function in mouse hippocampal neurons. We demonstrated that Tat (100 nM for 10 min) induced endolysosome de-acidification (Figure 1C). Endolysosome de-acidification can lead to the abnormal accumulation of undegraded materials, and subsequently to endolysosome enlargement [60,67]. We therefore measured changes in endolysosome size and demonstrated that Tat (100 nM for 48 h) significantly increased the size of the endolysosomes (Figure 1D). It has been shown that endolysosome dysfunction in neurons affects dendritic spine dynamics [33,68], and thus Tat-induced endolysosome dysfunction could lead to the observed dendritic spine impairments.

### 3.2. 17αE2 Prevents Tat-Induced Dendritic Damage

As the predominant form of estrogen in the brain [69,70], 17αE2 has been shown to promote dendritic spine and synapse formation [62,71]. Here, we evaluated the ability of 17αE2 to protect against Tat-induced dendritic damage. First, EGFP-expressing neurons pre-treated with 17αE2 (10 nM for 10 min) were treated with Tat (100 nM) for 10 min, and dynamic changes in the dendritic spines were monitored using time-lapse imaging. The same dendrite at 0 min (t-0) and 10 min (t-10) of Tat treatment was imaged. The net gain/loss in various types of spines over the 10 min treatment was calculated to assess dendritic spine turnover. We demonstrated that Tat treatment resulted in a rapid decrease in mushroom, stubby, and long/thin types of spines (Figure 2A). Pre-treatment with 17αE2 significantly prevented the Tat-induced decreases in mushroom and stubby types of spines, but not long/thin types of spines (Figure 2B). These findings show that 17αE2 rapidly modulates the plasticity of dendritic spines in primary hippocampal neurons and that 17αE2 blocks Tat-induced dendritic impairment. Next, we determined whether this neuroprotective effect of 17αE2 would persist over a longer period. We demonstrated that pre-treatment with 17αE2 (10 nM, 6h) significantly prevented Tat (100 nM for 48 h)-induced reductions in dendritic spine density (Figure 2C). 

### 3.3. 17αE2 Prevents Tat-Induced Endolysosome Dysfunction

Because dendritic spine remodeling is intrinsically linked to lysosomal function in neurons [34], we determined the ability of 17αE2 to prevent Tat-induced endolysosome dysfunction. We checked the levels of active cathepsin D (CatD) in neuronal endolysosomes (a functional outcome of endolysosome de-acidification) using the dye BODIPY FL-Pepstatin A, which binds to the active site of cathepsin D when its active site is exposed in an acidic environment. The proportion of active endolysosomes (active CatD positive) to total endolysosomes identified with LysoTracker Red (LTR) decreased upon the addition of Tat at 100 nM for 30 min (Figure 3). We demonstrated that 17αE2 pre-treatment (10 nM, 10 min) increased the percentage of active endolysosomes (CatD positive) and significantly prevented Tat-induced decreases in the percentage of active endolysosomes (CatD positive) (Figure 3). This is consistent with our previous findings that 17αE2 acidifies endolysosomes and enhances endolysosome function [62].

### 3.4. 17αE2 Enhances Endolysosome Function via ERα

In the hippocampus and hippocampal neurons, where the nuclear presence of estrogen receptors (ERs) is sparse [72,73,74,75], extranuclear membrane-bound ERs have been implicated in the enhancing effects of estrogen on cognition and synaptic function [76,77,78,79,80,81]. Such extranuclear membrane-bound ERs have been shown to have distinct subcellular distribution, with ERα expression on endolysosomes [75,82], ERβ on mitochondria [74,83,84], and G-protein coupled estrogen receptor 1 on the endoplasmic reticulum [85]. We have consistently demonstrated that that ERα is primarily expressed on endolysosomes, with ERα co-localizing with Rab7- and LAMP1-positive endolysosomes in hippocampal neurons [62]. Thus, it is possible that 17αE2 could activate endolysosome-localized ERα and initiate endolysosome-dependent actions.

To explore such a possibility, we first knocked down the expression of ERα with siRNA in mouse hippocampal neurons (CLU199) and observed a 70% reduction in ERα protein levels (Figure 4A). We then measured endolysosome pH using a ratio-metric method which combines the use of the pH-sensitive pHrodo-dextran and the pH-insensitive Texas-Red-dextran. We demonstrated that the endolysosome pH of ERα knockdown (ERαKD) neurons was significantly increased compared with that of scrambled siRNA-treated neurons (ERα scr) (Figure 4B). Given that 17αE2 can be endogenously produced by neurons [86,87], our findings suggest ERα knockdown could impair the endolysosome-enhancing effects of endogenously produced 17αE2. Furthermore, we demonstrated that exogenously added 17αE2 was less able to induce endolysosome acidification in ERα KD than in ERα scr cells (Figure 4C,D). Together, our findings suggest that ERα is necessary for the endolysosome-enhancing effects of 17αE2.

### 3.5. 17αE2 Protects against Tat-Induced Endolysosome Dysfunction and Impairment in Dendritic Spines via ERα

Having shown that 17αE2 acidifies endolysosomes via ERα, we explored the extent to which ERα knockdown affects the protective effects of 17αE2 against Tat-induced endolysosome dysfunction. Here, endolysosome function was assessed using alterations in the percentage of active CatD-positive endolysosomes in both ERα scr and ERα KD CLU199 cells (Figure 5A). Pre-treatment with 17αE2 significantly prevented Tat-induced reduction in the percentage of active endolysosomes (CatD-positive) in ERα scr cells, but not in ERα KD cells (Figure 5B). Our findings suggest that 17αE2 protects against Tat-induced endolysosome dysfunction via ERα.

Because ERα is present in dendritic spines [79], and because lysosomal activity and its mobility along the dendrites is involved in dendritic spine dynamics [32], we next determined whether ERα knockdown affects the protective effects of 17αE2 against Tat-induced dendritic spine impairment. Here, we measured the dendritic spine density in ERα scr and ERα KD neurons treated with Tat (100 nM for 30 min) in the presence or absence of 17αE2 pre-treatment (Figure 5C). We demonstrated that 17αE2 pre-treatment resulted in significantly increased dendritic spine density in Tat-treated neurons, but only in neurons treated with scrambled siRNA (ERα scr), not in ERα knockdown (ERα KD) neurons (Figure 5D). Thus, our findings suggest that ERα mediates the protective effects of 17αE2 against Tat-induced dendritic spine impairment. Furthermore, ERα-mediated endolysosome enhancement could underlie the protection afforded by 17αE2 against Tat-induced dendritic impairments.

### 3.6. 17αE2 Protects against Tat-Induced Endolysosome Dysfunction and Impairment in Dendritic Spines via Endolysosome Localization of ERα

We have shown that ERα is localized predominantly on Rab7-postive endolysosomes and, to a lesser extent, on LAMP1-positive endolysosomes [62]. Given that the membrane localization of ERα depends on palmitoylation [88,89,90], we generated a mouse-specific ERα mutant (C451A) which lacks the known palmitoylation site. We found that this ERα mutant failed to localize on Rab7-positive endolysosomes in mouse neuronal cells [62]. Furthermore, we demonstrated that 17αE2 exerts its enhancing effect on endolysosomes via endolysosome-localized ERα [62].

Because we had demonstrated that ERα is essential for the protective effect of 17αE2 against Tat-induced endolysosome dysfunction and damage of dendritic spines, we determined further whether the endolysosome localization of ERα mediates the above mentioned protective effects of 17αE2. To investigate whether the endolysosome localization of ERα mediates the protective effect of 17αE2 against Tat-induced endolysosome dysfunction, we overexpressed ERα C451A and determined the percentage of active endolysosomes (CatD-positive) in CLU199 mouse neuronal cells treated with Tat (100 nM for 30 min), both in the absence and presence of 17αE2 pre-treatment (Figure 6A). Because we have demonstrated that 17αE2 treatment increases the localization of ERα on endolysosomes [62], we reasoned that the over-expressed ERα C451A mutant, which fails to localize on endolysosomes, would compete the binding of 17αE2 to WT ERα. As such, the over-expressed ERα C451A mutant would block the 17αE2-induced localization of WT ERα to the endolysosomes, and thus attenuate the enhancing effect of 17αE2 on endolysosomes. As expected, we demonstrated that in cells expressing wild-type ERα (ERα WT), pre-treatment with 17αE2 increased the percentage of CatD-positive active endolysosomes when challenged with Tat (Figure 6B). In contrast, in cells overexpressing ERα C451A, 17αE2 pre-treatment failed to increase the percentage of active endolysosomes (CatD-positive) when challenged with Tat (Figure 6B).

To further explore whether the endolysosome localization of ERα contributes to the protection afforded by 17αE2 against Tat-induced dendritic damage, we measured the density of the dendritic spines in the ERα C451A over-expressing primary hippocampal neurons that were treated with Tat, both with and without 17αE2 pre-treatment. As with our earlier results with ERα KD, over-expressing the ERα C451A mutant impaired the ability of 17αE2 to increase the density of the dendritic spines in the Tat-treated neurons (Figure 6C,D). Thus, our findings suggest that ERα localized on endolysosomes is critical for 17αE2-mediated protection against Tat-induced endolysosome dysfunction and impairment in dendritic spines.

## 4. Discussion

The present study explores the protective role of 17αE2 against HIV-1 Tat-induced endolysosome dysfunction and dendritic impairment. The prominent findings of the present study are that Tat induces endolysosome dysfunction and reduction in dendritic spines, that 17αE2 protects against Tat-induced endolysosome dysfunction and reduction in dendritic spines, and that ERα and its endolysosome localization mediates the protective effects of 17αE2.

Endolysosomes, which include endosomes, lysosomes, and autolysosomes, form a dynamic and interconnected network inside the cell. A hallmark feature of endolysosomes is their acidic luminal environment [91,92,93,94], which is critical for the proper function of up to 60 pH-sensitive hydrolytic enzymes [95]. As terminal degradation centers, endolysosomes mediate the degradation of extracellular materials internalized by endocytosis and/or phagocytosis, as well as intracellular components delivered to lysosomes via autophagy. The acidic luminal pH is also critical for the sorting and trafficking of molecules and/or membranes to their proper destinations for proper processing and physiological functions [96,97,98]. 

Endolysosomes are especially critical for the proper function of neurons because neurons are long-lived and post-mitotic cells with extreme polarity and extensive processes. As long-lived and post-mitotic cells, neurons cannot remove non-degraded or partially degraded products and potentially toxic constituents through cell division, and thus quality control over a large volume of cytoplasm and membrane relies on an efficient proteolytic system, such as that of endolysosomes, with proteasomes and autolysosomes responsible for quality control over cytosolic macromolecules and organelles and endolysosomes for plasma membrane components [30,31]. Furthermore, neurons are extremely polarized cells, and they undergo extensive processes, complex dendritic arbors representing the information input centers and axons with lengths ranging from tens to hundreds of centimeters representing the information output center [99]. Such extreme polarity of neurons makes it challenging to regulate the trafficking of membrane components, which requires highly dynamic endolysosomes [35,100,101,102,103,104]. It is not surprising that mutations to endolysosomal genes preferentially affect the nervous system and contribute to neurodegenerative diseases [31,105,106]. Indeed, endolysosome dysfunction has been said to play a central role in a range of age-related neurodegenerative disorders [106,107,108].

Endolysosome dysfunction has been implicated by post-mortem brain tissues from HAND patients [26,27,28,29]. Experimentally, we and other have shown that various HIV-1-related factors, such as HIV-1 [109,110], HIV-1 proteins (including Tat [60,61,111], gp120 [112,113,114], Nef [115], and Vpr [116]), a subset ART drugs [117], morphine [64], and methamphetamine [118,119], induce endolysosome dysfunction. As a secreted HIV protein that is not reduced or blocked by current ART drugs [43,44,45], Tat is present on neurons [46,47] and is known to enter neurons via endocytosis [56]. Thus, internalized Tat could affect endolysosomes directly [60,61]. We have demonstrated that exogenously added Tat de-acidifies endolysosome pH, reduces the activity of endolysosome enzymes, and dramatically increases the size of endolysosomes. Such morphological and functional changes in endolysosomes could result from Tat-induced endolysosome de-acidification, which not only impairs the degradation ability of endolysosomes (leading to an abnormal accumulation of undegraded materials [60,61]), but also leads to the impaired fusion [120] and/or trafficking of endolysosomes [113,121]. Endolysosomes contribute to membranes and the transportation of membrane proteins to synapses and dendritic spines. Endolysosomes also play an important role in the degradation of dendritic cargo, and it not surprising that endolysosomes play a critical role in the modulating of the dynamics of dendritic spines and synaptic plasticity [32,33,34,35]. As such, Tat-induced endolysosome de-acidification and dysfunction could reduce their degradative capacity and impair their trafficking along dendrites [122], thus disrupting dendritic spine remodeling and leading to a reduction in dendritic spine density (Figure 7).

Given that endolysosome dysfunction contributes to dendritic impairment, enhancing endolysosome function and/or alleviating endolysosome dysfunction represents a therapeutic strategy against Tat-induced dendritic spine impairment. As the major form of estrogen in the brain [71,72], 17αE2 promotes dendritic spine and synapse formation [62,71]. Significantly, we have recently shown that the neuroprotective effect of 17αE2 depends, at least in part, on its enhancing effect on endolysosomes [62]. In the present study, we established that 17αE2 prevented Tat-induced reductions in dendritic spines. Based on our recent findings that 17αE2 enhances endolysosome function and increases the density of dendritic spines via endolysosome-localized ERα [62], we further determined whether ERα plays a role in the protective effect of 17αE2 against Tat-induced endolysosome dysfunction and impairment in dendritic spines. We demonstrated that siRNA knockdown of ERα or over-expressing an ERα mutant (C451A in mice) that fails to localize on endolysosomes impairs the protective effects of 17αE2 against Tat-induced endolysosome dysfunction and impairment in dendritic spines. Thus, our findings demonstrate clearly that 17αE2 can protect against Tat-induced direct synaptodendritic damage. In neurons, endolysosomes are responsible for transporting cargo both towards and away from the dendritic spines, their activity is required for the remodeling of dendritic spines [32,33], and ERα is localized in dendritic spines [79]. Thus, we speculated that activating endolysosome-localized ERα would mediate the endolysosome-acidifying effect of 17αE2 (Figure 7). Such endolysosome-acidifying effects of 17αE2 could lead to enhanced proteolysis and the proper travelling of endolysosomes along dendrites, which could in turn result in increased dendritic spine formation. However, our findings do not exclude the possibility that ERα localization on other membranes could also affect dendritic spine formation, and this warrants further investigation. 

In summary, our findings demonstrate clearly that Tat induces endolysosome dysfunction and reduction in dendritic spines, that 17αE2 protects against Tat-induced endolysosome dysfunction and impairment in dendritic spines, and that ERα and its endolysosome localization mediates the protective effects of 17αE2. Neurons are highly polarized cells, and the compartmentalized signaling of neurons plays a critical role in the formation and maintenance of dendritic spines and synaptic plasticity. Thus, our findings provide novel insights into the neuroprotective effects of 17αE2 which may lead to the development of new adjunct therapeutics against HAND.

## Figures and Tables

**Figure 1 cells-12-00813-f001:**
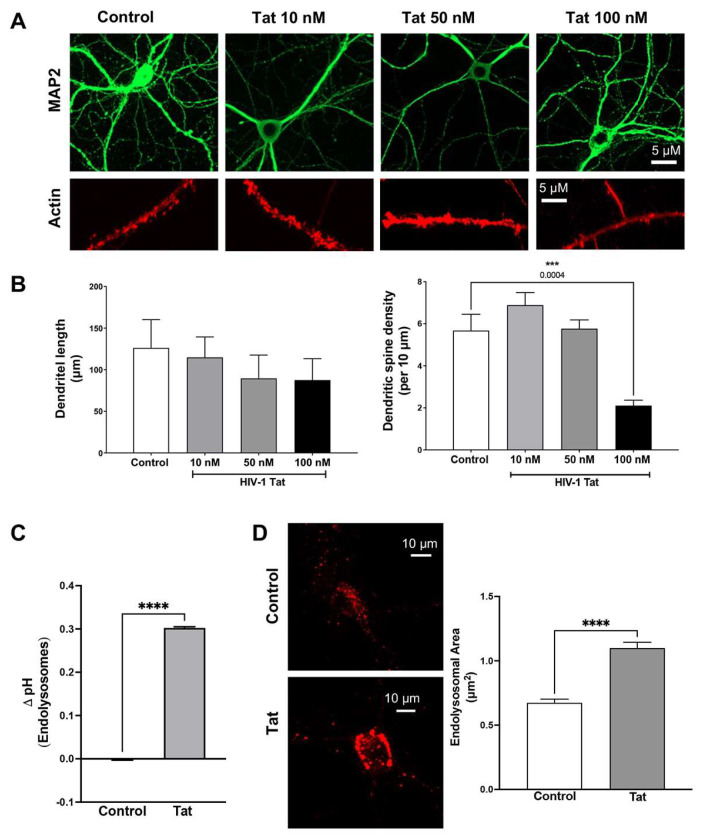
HIV-1 Tat induces dendritic impairment and endolysosome dysfunction. (**A**) Effects of Tat (10–100 nM for 48 h) on the morphology of dendrites (MAP2) and dendritic spines (phalloidin-actin) with heat-inactivated Tat as a control. (**B**) The quantitative data of (**A**) show that Tat (100 nM for 48 h) slightly decreased dendritic length and significantly reduced the density of dendritic spines (*n* = 15–30 neurons from 2 repeats, *** *p* < 0.001). (**C**) Tat (100 nM for 10 min)-induced endolysosome de-acidification (*n* = 3 repeats, **** *p* < 0.0001). (**D**) Tat (100 nM for 48 h) increased the size of the endolysosomes, as determined using LysoTracker Red (*n* = 15 neurons from 2 repeats, **** *p* < 0.0001).

**Figure 2 cells-12-00813-f002:**
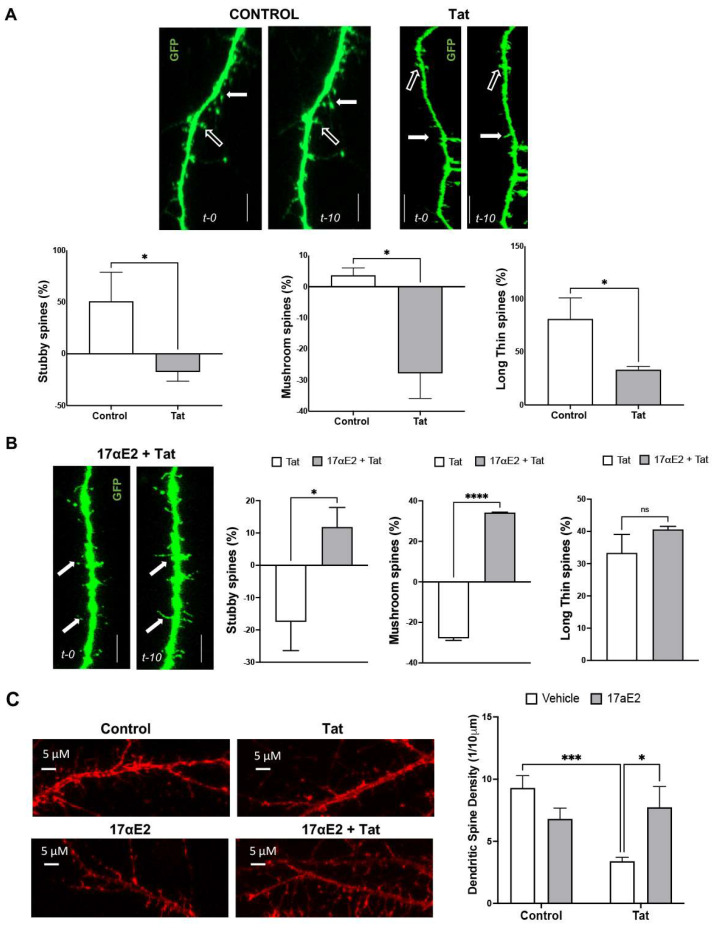
17αE2 protects against HIV-1 Tat-induced dendritic impairment. (**A**) Dendritic spine turnover in EGFP-expressing neurons. Spine growths are indicated by solid arrows, and spine reductions by hollow arrows (Scale = 5 µm). The quantitative data show dendritic spine turnover between 0 and 10 min. Spine formation is indicated by positive values, while spine elimination is indicated by negative values. Tat treatment (100 nM) results in a reduction in stubby, mushroom, and long/thin types of dendritic spines (*n* = 15, 180 neurons, * *p* < 0.05). (**B**) In the presence of Tat (100 nM), 17αE2 (10 nM) increases Tat-induced reductions in stubby, mushroom, and long/thin spines (*n* = 15, 180 neurons, * *p* < 0.05). (**C**) Representative confocal images of dendritic spines (phalloidin-actin). The quantitative data show that 17αE2 (10 nM, pre-treatment of 6 h) prevented a Tat (100 nM for 48 h)-induced reduction in total dendritic spine density (*n* = 3, 20 neurons, * *p* < 0.05, *** *p* < 0.001, **** *p* < 0.0001).

**Figure 3 cells-12-00813-f003:**
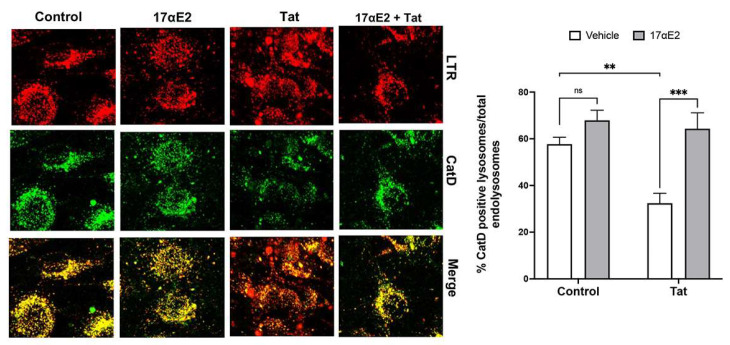
17αE2 prevents HIV-1 Tat-induced endolysosome dysfunction. The images and quantitative data show alterations in the percentage of active endolysosomes (active CatD, green) vs. total endolysosomes (LysoTracker, red). 17αE2 (10 nM, pre-treatment for 10 min) increased the percentage of active endolysosomes and prevented Tat-induced decreases in the percentage of active endolysosomes (*n* = 3, 20–30 neurons, ** *p* < 0.01, *** *p* < 0.001).

**Figure 4 cells-12-00813-f004:**
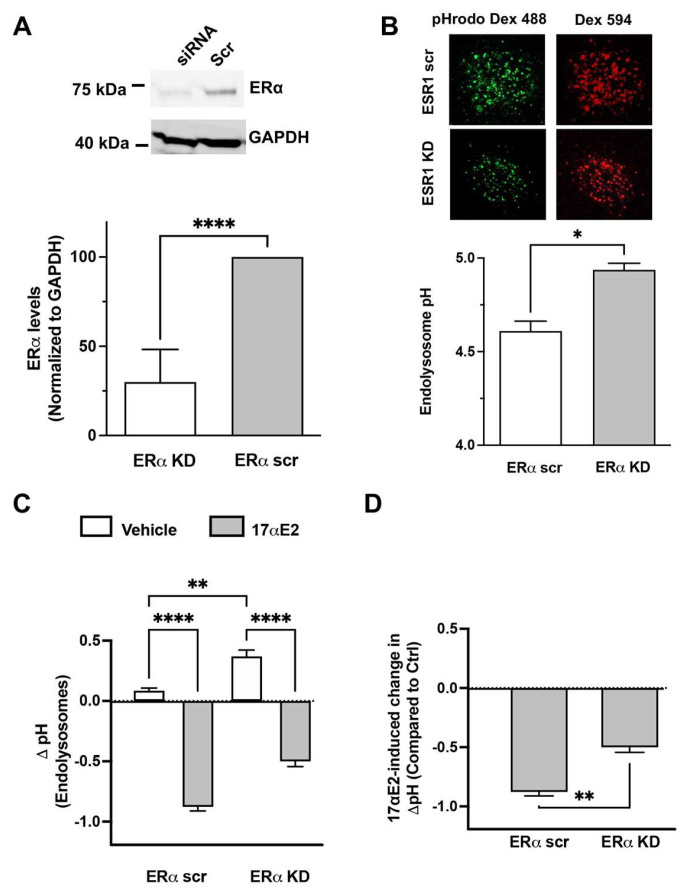
ERα knockdown attenuates 17αE2-induced acidifying effects on endolysosomes. (**A**) ERα protein levels were knocked down in a mouse hippocampal cell line (CLU-199) treated with ERα siRNA (*n* = 4, **** *p* < 0.0001). (**B**) ERα KD resulted in a greater de-acidification of endolysosomes compared with ERα scr (*n* = 3, * *p* < 0.05). Endolysosome pH was measured ratiometrically with the use of a pH-sensitive (pHrodo) and pH-insensitive (Texas Red) dextran. (**C**) 17αE2 (10 nM for 10 min)-acidified endolysosomes in ERα scr and ERα KD cells (*n* = 3, ** *p* < 0.01, **** *p* < 0.0001). (**D**) The magnitude of the 17αE2-induced decrease in endolysosome pH was significantly reduced in ERα KD cells (*n* = 3, ** *p* < 0.01).

**Figure 5 cells-12-00813-f005:**
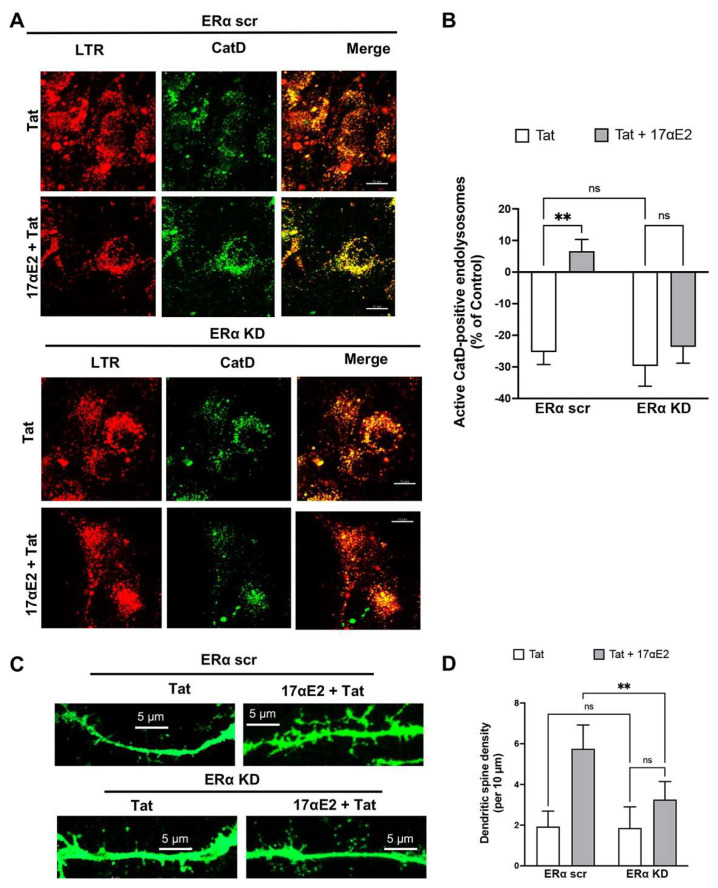
siRNA knockdown of ERα prevents the protective effects of 17αE2 against HIV-1 Tat-induced endolysosome dysfunction and dendritic impairment. (**A**) Tat (100 nM for 30 min)-induced alterations in the percentage of active endolysosomes (active CatD, green) vs. total endolysosomes (LTR, LysoTracker, red) in CLU199 cells treated with scrambled (scr) or targeted (KD) siRNA against ERα with or without 17αE2 (10 nM pre-treatment for 10 min). (**B**) In the presence of Tat (100 nM for 30 min), 17αE2 significantly increased the percentage of active endolysosomes in ERα scr cells, but not in ERα KD cells (*n* = 5, ** *p* < 0.01). (**C**) Tat (100 nM for 30 min)-induced changes in dendritic spines in EGFP expression neurons treated with scrambled siRNA (ERα scr) or siRNA against ERα (ERα KD). (**D**) In the presence of Tat (100 nM for 30 min), 17αE2 (10 nM, pre-treatment for 10 min) increased the density of dendritic spines in ERα scr neurons, but not in ERα KD neurons (*n* = 5, 20–30 neurons ** *p* < 0.01).

**Figure 6 cells-12-00813-f006:**
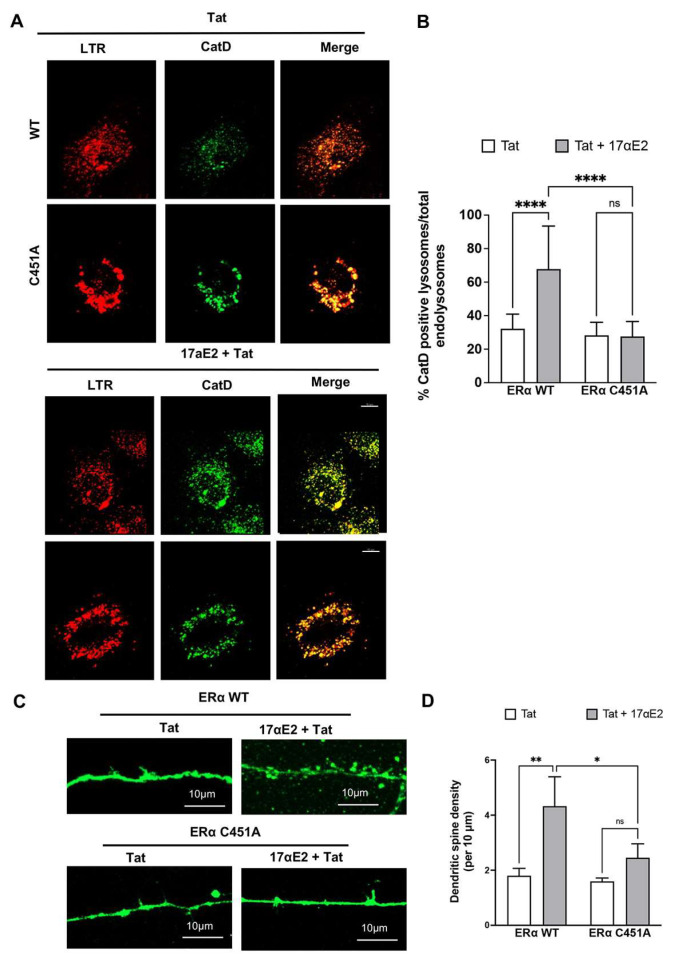
17αE2 protects against HIV-1 Tat-induced endolysosome dysfunction and impairment in dendritic spines via endolysosome-localized ERα. (**A**) Tat (100 nM for 30 min)-induced alterations in the percentage of active endolysosomes (active CatD, green) vs. total endolysosomes (LTR, LysoTracker, red) in CLU199 neuronal cells expressing wildtype ERα (ERα-HA) or ERα C451A-HA (ERα C451A) with and without 17αE2 (10 nM pre-treatment for 10 min). (**B**) In the presence of Tat (100 nM for 30 min), 17αE2 significantly increased the percentage of active endolysosomes in wild type cells, but not in ERα C451A over-expressing cells (*n* = 2, 11–20 neurons **** *p* < 0.0001). (**C**) Tat (100 nM for 30 min)-induced changes in dendritic spines in BacMAM EGFP-transduced wildtype (ERα-HA) neurons and ERα C451A-HA over-expressing neurons. (**D**) In the presence of Tat (100 nM for 30 min), 17αE2 (10 nM, pre-treatment for 10 min) increased the density of dendritic spines in wildtype (WT) neurons, but not in ERα C451A over-expressing neurons (*n* = 5, 20–30 neurons, * *p* < 0.05, ** *p* < 0.01).

**Figure 7 cells-12-00813-f007:**
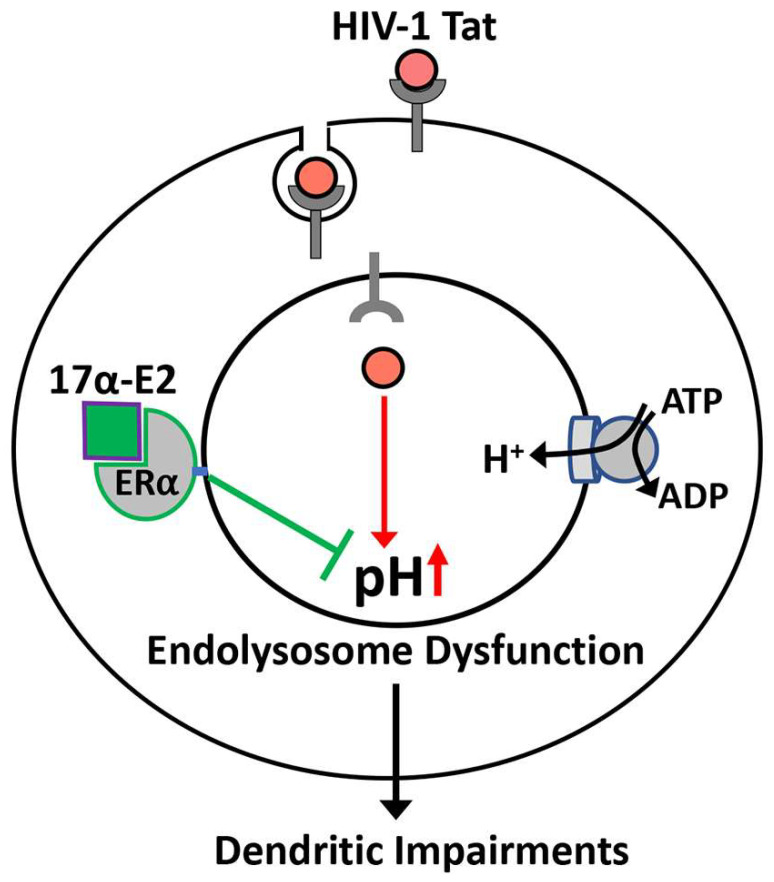
Proposed model according to which 17αE2 protects against HIV-1 Tat-induced endolysosome dysfunction and impairment in dendritic spines via endolysosome-localized ERα. Internalized Tat-induced endolysosome de-acidification and dysfunction could reduce their degradative capacity and impair their trafficking along dendrites, thus disrupting dendritic spine remodeling and leading to a reduction in dendritic spine density. Such damaging effects of Tat can by attenuated by 17αE2, which acidifies endolysosomes by activating endolysosome-localized ERα.

## Data Availability

The data presented in this study are available in this article.

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
