# Peer review of "17⍺-Estradiol Protects against HIV-1 Tat-Induced Endolysosome Dysfunction and Dendritic Impairments in Neurons"

_cells, 2023, doi:10.3390/cells12050813_

Round 1

Reviewer 1 Report

In the manuscript “17a-estradiol protects against HIV-1 Tat-induced endolysosome dysfunction and dendritic impairments in neurons” the authors experimentally demonstrate that 17aE2 protects neuronal endolysosomes damaging caused by HIV-1 Tat through different experiments

I have to say that I think the manuscript was well planned and is very clear, and is relevant on the field.

I just have one question; how did you measure the dendritic spine density?

On another hand, could you discuss a bit why HIV-1 Tat 10nM and 50nM concentrations hadn´t important effects as 100 nM? In figure 1B, it seems that when you used Tat 10nM the dendritic spine density is even more than control, and 50 nM seems to be similar than the control. Apart of this, it is very clear why you chose 100 nM for the next experiments.

At last, with your interesting results, I think It would be great if you have a propose model/diagram of your findings.

Author Response

Reviewer #1: In the manuscript “17a-estradiol protects against HIV-1 Tat-induced endolysosome dysfunction and dendritic impairments in neurons” the authors experimentally demonstrate that 17aE2 protects neuronal endolysosomes damaging caused by HIV-1 Tat through different experiments. I have to say that I think the manuscript was well planned and is very clear, and is relevant on the field. 

I just have one question; how did you measure the dendritic spine density?

Response: The method used for measuring the dendritic spine density in fixed neurons has now been added to the Methods section (page 3, line 105-114).

On another hand, could you discuss a bit why HIV-1 Tat 10nM and 50nM concentrations hadn´t important effects as 100 nM? In figure 1B, it seems that when you used Tat 10nM the dendritic spine density is even more than control, and 50 nM seems to be similar than the control. Apart of this, it is very clear why you chose 100 nM for the next experiments.

Response: The review’s observation is right. In Figure 1B, Tat at 10 nM slightly increased dendritic spine density, but the difference was not statistically significant. Similarly, Tat at 50 nM did not induce significant changes in dendritic spine density differences. This is partially because we are assessing the effect of Tat on total density of dendritic spines, which are the sum of mushroom, stubby, long/thin types of dendritic spines. We did observe that Tat slightly increased the density of long/thin type of spines, while concomitantly decreasing the density of the more mature stubby and mushroom types of spines. At lower concentrations, Tat induces relatively more increases in the density of long/thin type of dendritic spines, which contributes to the observed slight increases in the density of total dendritic spines by Tat at the concentration of 10 nM.

At last, with your interesting results, I think it would be great if you have a proposed model/diagram of your findings.

Response: As suggested, we have included a proposed model (Figure 7).

Reviewer 2 Report

This is a well written manuscript that evaluates the protective role of 17a-estradiol against HIV-1 Tat-induced endolysosome dysfunction and neuronal injury. The manuscript is a carefully prepared genetic validation of the involvement of estrogen receptor alpha and its endolysome localization to explain how 17a-estradiol mediates its protective effect.  The authors supplied compelling experimental evidence to support their hypothesis, the data were cross-validated using several genetic methodologies and the experiments were well designed with proper control groups and appropriate power. This reviewer does not have any suggestions to further improve on this excellent work.

Author Response

The authors thank the reviewer for the highly positive feedback.